# Olive Mill Pomace Extract Loaded Ethylcellulose Microparticles as a Delivery System to Improve Olive Oils Oxidative Stability

Filipa Paulo [1], Loleny Tavares [2,*] and Lúcia Santos [1,3]

[1]  Laboratory for Process Engineering, Environment, Biotechnology and Energy (LEPABE),
   Faculty of Engineering, University of Porto, Rua Dr. Roberto Frias, s/n, 4200-465 Porto, Portugal
[2]  School of Design, Management and Production Technologies Northern Aveiro (ESAN), University of Aveiro,
   Estrada do Cercal 449, 3720-509 Oliveira de Azeméis, Portugal
[3]  Associate Laboratory in Chemical Engineering (ALiCE), Faculty of Engineering, University of Porto,
   4200-465 Porto, Portugal
*  Correspondence: tavaresloleny@ua.pt or tavaresloleny@gmail.com

**Abstract:** The protective effect of olive mill pomace (OMP) loaded ethylcellulose microparticles as an alternative to synthetic antioxidants against the oxidation of olive oils was assessed. OMP extract was obtained by an optimized two-step solid-liquid extraction; encapsulation was performed by double emulsion solvent evaporation technique considering a theoretical loading content in phenolic compounds of 5% ($w/w$). The changes in the peroxide values, the *p*-anisidine values, the total oxidation values, the free fatty acids content, the total antioxidant activity, and the total phenolic content were synchronized under storage at 62 °C. The results of oxidative stability were compared with plain oils, oils enriched with synthetic antioxidants, and oils fortified with OMP extract. The encapsulation efficiency of phenolic compounds was 96.0 ± 0.3%. The fortification of olive oils with microparticles retarded the appearance of peroxides, reduced the content of secondary oxidation products, and slowed down hydrolysis processes. The microparticles were efficiently designed to sustain the release of antioxidants to control the oxidative status of oil samples, retarding the free fatty acids formation rather than synthetic antioxidants. The results of this study bring new perspectives regarding the potential use of encapsulated extracts rich in antioxidants as an alternative to synthetic antioxidants to improve oil oxidative stability.

**Keywords:** olive oils; fortification; oxidative stability; microparticles; peroxides; phenolic compounds; antioxidants; olive mill pomace





## 1. Introduction

Olive oil from olive fruit (*Olea europaea* L.; *Oleaceae*) is an outstanding source of a great variety of bioactive compounds such as monounsaturated free fatty acids (MUFA) (e.g., oleic acid), hydrocarbon squalene, aroma compounds, tocopherols, and phenolic compounds [1,2]. A hallmark of the Mediterranean diet is the consumption of olive oil as an essential fat, covering up to 17 to 25% of the typical Mediterranean diet calories per day, which has been associated with a low incidence and prevalence of cardiovascular diseases, diabetes type 2, some neurodegenerative diseases (stroke, Alzheimer's and Parkinson's diseases) and certain types of cancer (e.g., prostate, colon and breast cancers) [3–5]. It has been acknowledged that the most significant factors affecting oil quality attributes and its shelf-life are lipid oxidation, which drives undesirable changes in texture, flavor, odor, and taste [3,6]. The nutritional losses associated with lipidic oxidation are associated with reducing consumer compliance and industrial economic losses. In this context, the olive oil industry has been devoted to diminishing lipid oxidation and improving olive oil stability [7]. Synthetic antioxidants, namely butylated hydroxytoluene (BHT), butylated hydroxyanisole (BHA), and even *tert*-butyl hydroquinone (TBHQ), have been extensively used as food additives to overcome issues regarding the stability of oils and fats [8,9].

Even though these synthetic antioxidants are generally recognized as safe (GRAS) and have been used as food additives for years, the demand for natural antioxidants has recently increased due to the potential toxicity and carcinogenicity commonly attributed to synthetic compounds [10]. Therefore, regarding safety concerns, food expert investigations have been focusing on the replacement of these synthetic antioxidants with natural ones. Naturally-occurring antioxidants may retard oxidative rancidity (i) through the scavenging of oxygen molecules, (ii) by capturing free radicals, or (iii) by deoxidizing/decomposing peroxides [10,11]. Among natural-occurring antioxidants, plant phenolic antioxidants have been the highlight of lipid oxidation prevention and health-promoting properties [12]. Olive mill pomace (OMP) is considered a relevant source of phenolic compounds with biological properties [13,14]. However, many phenolic compounds are sensitive to environmental conditions such as pH, light, temperature, oxygen, moisture content, etc. [3,13]. Therefore, their encapsulation is a straightforward technological strategy to protect these valuable compounds embedded in food matrices prone to rancidity as oils (e.g., olive oil) [13,15]. OMP is a natural source of phenolic compounds to effectively stabilize oils such as olive oil through their encapsulation and controlled release [16]. Encapsulation can protect and increase phenolic compounds properties (e.g., stability, solubility, bioavailability, and antioxidativity), and simultaneously control the release of these bioactive compounds to efficiently retard lipidic oxidation in rancidity-prone food matrices such as vegetable oils [17,18]. Among the encapsulation techniques, the $w_1/o/w_2$ water-in-oil-in-water double emulsion solvent evaporation technique is an attractive method to protect water-soluble compounds susceptible to environmental degradation as phenolic compounds. For instance, these multicompartmental microsystems are obtained through the preparation of a primary emulsion ($w_1/o$), which is re-emulsified in an external aqueous phase ($w_2$) containing appropriate emulsifiers (e.g., polyvinyl alcohol—PVA) [3,15,16,19,20]. Among the coating materials used for encapsulation proposes by the $w_1/o/w_2$ double emulsion solvent evaporation technique, ethylcellulose is a biocompatible cellulose-derived class of polymers attractive to food applications, and their application as carriers in food additives is approved by the European Union (European Commission Regulation 1130/2011) [21,22]. Ethylcellulose polymers (differing on the ethylation degree) have been receiving attention as they are gastrointestinal resistant polymer-carriers, are tasteless, odorless, non-toxic, non-irritant, and also, they are stable to many environmental conditions such as heat, moisture, light, and oxygen. Moreover, they present outstanding resistance to mechanical stress [22,23].

The beneficial effects of olive oil are associated with various classes of bioactive components mainly monounsaturated and polyunsaturated fatty acids, squalene, triterpenic acids, phytosterols, dialcohols, tocopherols, and polyphenols which have strong antioxidant activity [24]. However, during olive oil extraction processes, many bioactive compounds, mainly phenolic compounds, can be retained in the olive byproducts essentially in olive pomace, olive leaves, and olive mill wastewater [22]. Additionally, the food industries use olive oil products as an ingredient in a variety of applications in addition to being used by the end consumer as a flavoring and cooking fat [25]. In this sense, to the best of our knowledge, no work is available addressing the effectiveness of OMP extract-loaded ethylcellulose microparticles as a delivery system to improve olive oil oxidative stability. Therefore, the purpose of this study was to evaluate the effectiveness of encapsulated olive mill pomace extract on the retarding lipidic oxidation of three types of olive oil, extra-virgin, virgin, and a blend of refined and virgin olive oils, compared to the embedment in lipidic matrices of only olive mill pomace extract and synthetic antioxidants (BHA and BHT considered alone or in a blend) using the Schaal oven test. Over a 24-day study at 62 °C, olive oil quality and nutritional values were evaluated by monitoring peroxide values, *p*-anisidine values, total oxidation values, free fatty acids content, total antioxidant activity, and total phenolic content changes over storage time.

## 2. Materials and Methods

### 2.1. Materials

The OMP samples were acquired from a local olive oil mill in Portugal (Vilas Boas, Vila Flor, Bragança, 41359822, −7123743). Hydrochloric acid solution at 37% *v/v* (EMSURE® ACS, Supelco®), hexane (HiPerSolv CHROMANORM®), ethanol (PESTINORM®) and ethyl acetate (SupraSolv®), chloroform (Supelco®, Ref: 02487), glacial acetic acid (Ref: 1.01830.2500), isooctane (Ref: PHR1915) and the *p*-anisidine reagent (Ref: A88255), phenolphthalein (Ref: 1.07233), sodium hydroxide (Ref: 1.06467.9010), methanol (Ref: 1.03726.2002) and dichloromethane (Ref. ACRO433991000) were obtained from VWR International (Fontenay-sous-Bois, France). Ethylcellulose (Ref: 433837-250G) and polyvinyl alcohol (PVA) (Ref: P8136-250G), gallic acid standard (Ref: 91215), Trolox standard (Ref: 238813), butylated hydroxyanisole (BHA) standard (Ref: B1253) and butylated hydroxytoluene standard (BHT) (Ref: W218405), potassium iodide (Ref: 221945), sodium thiosulphate (Ref: 217263, $Na_2S_2O_3$), and starch (Ref: 33615), Folin–Ciocalteu reagent (Ref: 47641-500ML-F), 2,2-diphenyl-1-picrylhydrazyl (Ref: D9132), and sodium carbonate anhydrous (Ref: 1613757) were obtained from Sigma Aldrich Chemical (St. Louis, MO, USA). The water used in this work was de-ionized and double-distilled using a Millipore™ water purification system (Burlington, Massachusetts (MA), USA) having 18.2 Ω electrical resistivity. Three alternative olive oils were considered in the present study: (i) a commercial extra-virgin olive oil (EVOO), (ii) a virgin olive oil (VOO), and (iii) a commercial blend of refined olive oil and virgin oil (ROO) were obtained from a local market in the city of Porto in Portugal.

### 2.2. Extraction and Encapsulation of Phenolic Compounds from Olive Mill Pomace

The OMP samples were obtained and stored at −22 °C (moisture content of 71.0 ± 5.5% *w/w*) prior to analyses. The OMP samples were freeze-dried on a benchtop freeze-dryer (SP Scientific, Stone Ridge, NY, USA) for 72 h. Dried samples were then grounded on an electric mill (Qilive Q5321 Grinder) to an average particle size of 142.2 ± 9.6 μm. Prior to phenolics extraction, the OMP sample (1 g) was submitted to a pre-treatment that included (i) an acidic hydrolysis handling and (ii) a fat removal step, according to the method reported by Paulo, Tavares and Santos [15]. The (i) acidic hydrolysis was performed by admixing the sample with 25 mL of an aqueous solution of HCl (0.1 M, pH 2) in a 50 mL Erlenmeyer flask, continuously shaken in an orbital shaker for 12 h. The acidic hydrolysis prompted the break of both ester and glycosylic bonds. The fats (ii) were removed, adding 5 mL/g of hexane to the hydrolyzed and ground OMP sample in a 50 mL Erlenmeyer flask. The final filtrate was admixed with ethyl acetate in a 50 mL PP centrifuge tube (extraction solvent volume corresponding to three times the final filtrate volume), vigorously shaken, and vortexed for 5 min. The mixture was then ultrasonicated for 15 min and centrifuged $2670\times g$, 15 min. Afterward, the supernatant was submitted to solvent evaporation using a rotary evaporator (BUCHI R-210, Buchi Laboratotiums Tchnik AG, Flawil, Switzerland) at 50 °C. Solvent traces were removed by a gentle nitrogen stream. The final crude dried extract was stored at −22 °C before prior analyses. The extraction experiments were performed in triplicate [15,16,23].

The obtained extract from the OMP sample was embedded into ethylcellulose microparticles by the water-in-oil-in-water ($w_1/o/w_2$) double emulsion solvent evaporation technique [15,16,23]. In the present study, a theoretical loading content (TLC) of 5% *w/w* in phenolic compounds was considered. The dried OMP extract was reconstituted in 3 mL of ultrapure water (UPW). Successive dilutions were conducted to achieve the desired loading content in phenolic compounds (theoretical loading content of 5% *w/w* corresponding to a concentration of 5.3 g of phenolic compounds/L of extract). The diluted sample from the OMP final extract constituted the internal aqueous phase ($w_1$). The organic phase (o) was formulated by admixing 100 mg of the polymer carrier—ethylcellulose—with 10 mL of dichloromethane to achieve a 10 g/L polymer concentration. The polymer solution was ultrasonicated for 15 min in an ultrasonic bath. Subsequently, to the polymer solution, 1 mL of the $w_1$ was added. The dispersed phase ($w_1/o$ emulsion) was vigorously shaken

and vortexed for 5 min. Then the $w_1/o$ emulsion was poured into 100 g of a PVA solution (1% $w/w$), and the mixture was emulsified using a high-performance liquid homogenizer at 5000 rpm for 5 min. The solvent evaporation from the mixture $w_1/o/w_2$ double emulsion was promoted through the continuous stirring of the $w_1/o/w_2$ double emulsion in a stirring plate at 700 rpm for 3 h, in the fume hood at room temperature ($20 \pm 2$ °C). Subsequently, microparticles were recovered by vacuum filtration using a 0.2 μm Whatman™ nylon membrane filter and washed using 500 mL of distilled water. Then, particles were collected, frozen for 24 h at $-22$ °C, and freeze-dried for 72 h. Encapsulation experiments were performed in triplicate.

### 2.3. Characterization of Olive Mill Pomace Extract-Loaded Ethylcellulose Microparticles

Olive mill pomace extract-loaded ethylcellulose microparticles were characterized by encapsulation efficiency and the actual loading content on (i) antioxidants and (ii) phenolic compounds. The encapsulation efficiencies were determined according to Equation (1), as described by Paulo, Tavares and Santos [15]:

$$EE_i\ (\%,\ w/w) = \frac{w_i}{w_{I,i}} \times 100 = \frac{w_{I,i} - w_{0,i}}{w_{I,i}} \times 100 = \frac{w_{I,i} - \left(w_{S,i} + w_{P,i}\right)}{w_{I,i}} \times 100 \qquad (1)$$

where $i$ denotes the cluster of compounds (e.g., antioxidants, phenolic compounds). The $w_i$ corresponds to the weight of compounds in the microparticles, which was assessed considering the weight of the bioactive compounds initially added for microparticle formulation ($w_{I,i}$) and the weight of non-encapsulated bioactives ($w_{0,i}$). The $w_{0,i}$ was assessed considering the weight of bioactives in the recovered supernatant of a sample of the double emulsion after 3 h of microparticles hardening ($w_S$) after the sample was centrifuged $2670\times g$, 15 min. The $w_P$ corresponds to the weight of bioactive compounds adsorbed on the surface of the microparticles. The $w_P$ was evaluated after submitting the multiple emulsion samples to low-intensity centrifugation. The pellet recovered from the centrifugation process was reconstituted in UPW and resubmitted to a centrifugation process for 15 min. The $w_P$ corresponded to the weight of compounds in the supernatant after the two-step centrifugation process. The EEs were evaluated in triplicate.

The actual loading content (*ALC*) was evaluated according to Equation (2) as presented by Paulo, Tavares and Santos [15].

$$ALC_i\ (\%,\ w/w) = \frac{w_i}{w_M} \times 100 = \frac{w_{I,i} - w_{0,i}}{w_M} \times 100 = \frac{w_{I,i} - \left(w_{S,i} + w_{P,i}\right)}{w_M} \times 100 \qquad (2)$$

where $w_M$ is the weight of microparticles recovered after freeze-drying. The *ALCs* were evaluated in triplicate.

### 2.4. Schaal Oven Storage Stability Test

The oil shelf-life, the period of time until the oil develops rancidity, is considered an utmost quality factor during the processing and marketing of vegetable oils. The oil shelf-life is assessed by its oxidative stability. Therefore, oxidative stability evaluation methods have been developed to evaluate oil shelf-life. Among the available oxidative stability evaluation methods, the Schaal oven test is a straightforward method that allows for assessing the rancidity state of oils. According to Schaal oven test experiments, elevated storage temperatures are employed to intentionally destroy the original physicochemical characteristics of oils [26,27]. In the present study, the Schaal oven test was considered to evaluate OMP extract-loaded ethylcellulose microparticles in retarding the oxidative deterioration of three types of extra-virgin olive oil, virgin olive oil, and refined olive oil. The enrichment of olive oil with BHA and BHT was performed up to the legal limit of 200 mg/kg of oil [21]. This reference value was considered during the addition of the extract and microparticles for olive oil fortification. Experiments were performed

considering (i) blank controls (oils without added antioxidants), (ii) positive controls (oils enriched with BHA, or BHT, or even a mixture of 50% BHA and 50% BHT to their legal limit of 200 mg/kg) and enriched samples (oils enriched with extract up to 200 mg/kg of oil and oils enriched with microparticles to their limit of 200 mg of antioxidants/kg of oil). Prior to the Schaal oven test experiments, compounds, or samples (BHA, BHT, BHA + BHT, extract, and microparticles) were precisely weighed in a glass beaker and solubilized in ethanol. The amount of ethanol used was limited, not surpassing 4% of the final oil weight, as described by Michotte, et al. [28]. The Schaal oven tests were performed as described by Yang, Song, Sui, Qi, Wang, Li and Jiang [27], with slight modifications. The controls (blank oils), the positive controls (oils plus BHA and BHT, individually or in combination), and olive oils incorporated with OMP extract and loaded microparticles were accurately weighed ($50 \pm 0.01$ g) in flasks with limited headspace. Prior to the beginning of the studies, samples were vortexed for 10 min and flushed with nitrogen for 3 min, as described by Michotte, Rogez, Chirinos, Mignolet, Campos and Larondelle [28]. Samples were stored in an oven at $62 \pm 2$ °C for 24 days. Samples were taken every 6 days for the determination of TAA, TPC, peroxide value (PV), *p*-anisidine value (*p*-AV), total oxidation (TOTOX) status, free fatty acids content, and quantification of the $K_{232}$ and $K_{270}$ extinction coefficients, which are indicators of olive oil stability. After oven removal, samples were flushed with nitrogen and stored at $-20$ °C until analysis. One flask per time point in triplicate was considered for each sample (controls, positive controls, oils fortified with extract, and oils enriched with microparticles).

*2.5. Determination of Oxidative Stability Indices*

2.5.1. Determination of Peroxide Value (PV)

The peroxide value (PV) of oil samples (blank controls, positive controls, oils enriched with OMP extract, and oils incorporating microparticles) was determined as described by Sun-Waterhouse, et al. [29]. In each experiment, 0.5 g of oil was dissolved in 3 mL of a solution of acetic acid/chloroform (3/2 *v/v*). Then, 50 μL of saturated potassium iodide (KI) solution was added, and the mixture was left to equilibrate for 1 min. Afterward, a volume of 3 mL of UPW was added. The mixture was then titrated with a 0.01 N standardized sodium thiosulphate solution until the yellow iodine color disappeared. Subsequently, 0.2 mL of starch indicator solution was added (10 g/L). The titration experiment was continuously processed until the blue color from the iodine disappeared. A PV experiment was performed using the blank sample as a control. Results are expressed as peroxide milliequivalents per kg oil (mEq/kg oil).

2.5.2. Determination of *p*-Anisidine Value (*p*-AV)

The *p*-AVs were determined as described by Sun-Waterhouse, Zhou, Miskelly, Wibisono and Wadhwa [29], with slight modifications. In each experiment, 0.5 g of oil were dissolved in 12.5 mL of isooctane. Isooctane was selected as the reference (blank). The oil mixture (or blank) (5 mL) was admixed with the *p*-anisidine reagent (1 mL at a concentration of 2.5 g/L in glacial acetic acid). The obtained mixture was vortexed for 10 min. An aliquot of the mixture (200 μL) was withdrawn to a 96-well microplate. The absorbance was then read using a microplate reader (Synergy HT, Biotek, Vermont (VT), USA) at 350 nm.

2.5.3. Determination of the Total Oxidation (TOTOX) Value

It has been described that a high rate of hydroperoxide generation is not necessarily associated with a high rate of generation of secondary oxidation products—high PVs do not imply that high *p*-AVs are observed [30,31]. In this regard, the total oxidation (TOTOX) value is a useful parameter that provides an overall picture of the oxidative status of the oil. The TOTOX values were calculated as described by Wai, Saad and Lim [31], as presented in Equation (3):

$$TOTOX \text{ value} = 2\,PV + AV \tag{3}$$

where *PV* is the peroxide value of the oil, and *AV* is the *p*-anisidine value of the oil.

### 2.5.4. Determination of Free Fatty Acids (FFA) Content

Most of the fatty acids in olive oil are long-chain fatty acids; indeed, it is described that olive oils present high contents of mono-saturated fatty acids (MUFA) and low contents of polyunsaturated fatty acids (PUFA) [32]. Nevertheless, long-chain fatty acids can be converted into short-chain fatty acids that may release free fatty acids (FFA) with time [29]. The liberated fatty acids may suffer β-oxidation, resulting in the formation of methyl ketones and aliphatic alcohols, in a process known as ketonic rancidity [33]. Therefore, the FFA quantification is considered a relevant quality control parameter of the oxidative stability of olive oils. The FFAs content was determined through direct titration. Accordingly, in a flask, an aliquot (1.25 mL) of ethanol, two drops of oil, and 50 µL of the phenolphthalein indicator were admixed, constituting the neutralized alcohol. The flask was then placed in a water bath at 60 °C. A solution of NaOH (0.01 N) was added to the neutralized alcohol until the appearance of a permanent faint pink color. Then, 1.41 g of the sample (oil) was added to the neutralized alcohol and titrated with NaOH until the appearance of the faint pink color, which corresponds to the endpoint of the phenolphthalein indicator.

The FFA content was expressed as oleic acid, as a percentage Equation (4):

$$\% \, FFA \, (\text{as oleic acid}) = \frac{V_{\text{NaOH}} \times N_{\text{NaOH}} \times 282.46}{W} \times 100 \tag{4}$$

where, $V_{\text{NaOH}}$ is the volume of NaOH titrant (mL), $N_{\text{NaOH}}$ is the normality of NaOH titrant (mol/1000 mL); the $W$ is the mass of the oil (g), and the value 282.46 corresponded to the molecular weight of oleic acid. Results are presented as mean ± standard deviation.

### 2.5.5. Changes on the $K_{232}$ and $K_{270}$ Extinction Coefficients

The extinction coefficients $K_{232}$ and $K_{270}$ are excellent indicators of the oxidative status of oil. The extinction coefficient values give some insights into the quality, preservation, and changes that occurred in the oily matrix through technological processes [34]. The $K_{232}$ and $K_{270}$ extinction coefficients were measured using a UV-Vis spectrophotometer (UV–VIS V-530 Jasco spectrophotometer, Oklahoma (OK), USA). The absorption values were obtained for a concentration of 1% $w/v$ in cyclohexane in a 10 mm cell [35].

### 2.5.6. Evaluation of the Total Antioxidant Activity and the Total Phenolic Content on Olive Oil Samples

Prior to determining the total antioxidant activity and the total phenolic content, oil samples were submitted to an extraction similar to the method described by Bail, et al. [36]. Briefly, in each experiment, 1 g of oil was extracted three times with 5 mL of a methanol/water (90/10 $v/v$) solution. The mixture was vortexed for 10 min and centrifuged $2670 \times g$ for 10 min. The methanolic extract was concentrated through the flush with nitrogen and stored at −20 °C until analysis. Before analyzing the total antioxidant activity (TAA) and total phenolic content (TPC), the dried extract was dissolved in a 10/90 methanol/water solution (1 mL).

The TAA was determined through the estimation of the free radical-scavenging ability (RSA) of samples using the 2,2-diphenyl-2-picrylhydrazyl radical (DPPH$^\bullet$) [37]. The inhibition percentage of DPPH$^\bullet$ discoloration (%$I$) was calculated based on Equation (5) as follows:

$$\% \, I = \frac{A_0 - A_1}{A_0} \times 100 \tag{5}$$

where $A_0$ is the absorbance of the control reaction (DPPH$^\bullet$ in methanol) and $A_1$ is the absorbance of the DPPH radical plus tested sample in methanol. The TAAs were expressed as milligrams per gram of dry OMP (mg/$g_{\text{OMP}}$) or as grams per liter of extract (g/L extract) or even as milligrams of Trolox equivalents (TE) per 100 mL of oil (mg$_{\text{TE}}$/100 mL oil).

The TPC was determined by the Folin–Ciocalteu micro-method, as described by Barroso, et al. [38], based on a scale-down procedure of the original method proposed by Singleton, et al. [39]. The TPC was expressed as milligrams of gallic acid equivalents (GAE)

per gram of dry OMP ($mg_{GAE}/g_{OMP}$) or as grams of gallic acid equivalents (GAE) per liter of extract ($g_{GAE}/L$ extract) or even as milligrams of gallic acid equivalents per kilogram of oil ($mg_{GAE}/kg$ oil).

### 2.6. Statistical Analysis

The experiments were performed in triplicate. Therefore, results are presented as mean ($n = 3$) $\pm$ standard deviation. One-way analysis of variance (one-way ANOVA) was used to determine significant differences. Values of $p < 0.05$ were considered statistically significant. Differences between means were analyzed by Tukey's test at a significance level of $p \leq 0.05$, using SAS (version 9.3) software.

## 3. Results and Discussion

### 3.1. Olive Mill Pomace Extract and Microparticles Characterization

The extract (n = 3) obtained after the extraction procedure presented a TAA of $79.1 \pm 7.9\%$ and a TPC of $50.5 \pm 1.5$ mg/$g_{OMP}$ (Table 1).

**Table 1.** Antioxidant activity and phenolic content of the olive mill pomace extract.

| Antioxidant Activity [a] | |
|---|---|
| $IC_{50}$ mg/$g_{OMP}$ | $223.9 \pm 1.2$ |
| mg/L extract | $74.6 \pm 0.4$ |
| **Phenols Content [a]** | |
| $mg_{GAE}/g_{OMP}$ | $50.5 \pm 1.5$ |
| $mg_{GAE}/L$ extract | $16.8 \pm 0.5$ |

[a] Results are presented as mean $\pm$ standard deviation ($n = 3$). GAE—Gallic Acid Equivalents; $IC_{50}$—Half-inhibitory concentration; OMP—Olive Mill Pomace.

Olive mill pomace-loaded ethylcellulose microparticles were successfully produced by water-in-oil-in-water ($w_1/o/w_2$) double emulsion solvent evaporation. Microparticles were formulated considering a TLC in phenolic compounds of 5% *w/w* (ALC of $4.8 \pm 0.9\%$). Results regarding the encapsulation efficiency of antioxidants and phenolic compounds are presented in Table 2.

**Table 2.** Main physicochemical characteristics of olive mill pomace loaded ethylcellulose microparticles.

| Bioactive Compounds | Encapsulation Efficiency (EE) (% *w/w*) | Theoretical Loading Content (TLC) (% *w/w*) | Actual Loading Content (ALC) (% *w/w*) |
|---|---|---|---|
| Antioxidants (AO) | $92.6 \pm 2.1$ | $18.9 \pm 2.7$ | $17.5 \pm 1.6$ |
| Phenolic Compounds (PC) | $96.0 \pm 0.3$ | $5.0 \pm 0.4$ | $4.8 \pm 0.9$ |

Results are presented as mean $\pm$ standard deviation ($n = 3$). ALC—Actual Loading Content; AO—Antioxidants; BAC—Bioactive Compound; EE—Encapsulation Efficiency; PC—Phenolic Compounds; TLC—Theoretical Loading Content.

Moreover, in Table 2, the results concerning the actual loading contents are presented compared to the theoretical loading contents considered in the present study. The encapsulation efficiencies varied from $92.6 \pm 2.1$ in the case of antioxidants and $96.0 \pm 0.3\%$ in the case of phenolic compounds. The majority of antioxidants and phenolic compounds present in the OMP extract were efficiently embedded into ethylcellulose microparticles. The actual loading contents were similar to the TLC considered, indicating the efficient incorporation of phenolic compounds into ethylcellulose microparticles. Microparticles were designed considering a TLC in phenolic compounds of 5% *w/w*. The obtained loading content in phenolic compounds was $4.8 \pm 1.2\%$. About $17.5 \pm 2.9\%$ of the weight of microparticles corresponded to antioxidants' weight. These are outstanding regarding the design of tailored-made microparticles encapsulating bioactive compounds extracted from agricultural wastes.

### 3.2. Changes in the Peroxide Values (PVs)

The peroxide values (PVs) that measure hydroperoxide concentration are the most employed chemical method to evaluate oils' oxidative deterioration. The PV measurements are based on the redox reaction between hydroperoxides and the excess of KI in an acidic medium, which results in the stoichiometric release of molecular iodine that is titrated against thiosulphate solution [31]. Even though the accuracy of PV is doubted as hydroperoxides decompose into a mixture of volatile and non-volatile products, they also react further to endoperoxides and other similar products. Therefore, PV is still a useful tool to monotonize the oxidative deterioration of oils [33]. The PV is commonly employed to assess the formation of hydroperoxides—the main initial products from oil oxidation [40]. Hydroperoxides are related to fatty acid susceptibility to oxidation and oil oxidative status [34]. It is a mandatory quality parameter that must be evaluated in olive oil commercialization [35]. The results regarding the PV levels of EVOO, VOO, and ROO during the 24-day storage are presented in Table 3.

**Table 3.** Changes in the peroxide value of olive oils during storage.

| Olive Oil Sample | Storage Time (Days) | Peroxide Value (mEq/kg) * | | | | | |
|---|---|---|---|---|---|---|---|
| | | Blank | BHA | BHT | BHA + BHT | OMP Extract | MPs |
| EVOO | 0 | 11.5 ± 0.2 [e] | 11.3 ± 0.2 [e] | 11.5 ± 0.3 [d] | 11.2 ± 0.4 [e] | 11.6 ± 0.1 [d] | 11.5 ± 0.2 [e] |
| | 6 | 15.5 ± 0.1 [d] | 13.4 ± 0.2 [d] | 14.5 ± 0.2 [c] | 13.9 ± 0.2 [d] | 13.1 ± 0.1 [c] | 12.1 ± 0.1 [d] |
| | 12 | 17.7 ± 0.1 [c] | 15.8 ± 0.2 [c] | 15.7 ± 0.1 [b] | 15.4 ± 0.2 [c] | 14.9 ± 0.2 [b] | 15.8 ± 0.2 [c] |
| | 18 | 18.6 ± 0.1 [b] | 17.8 ± 0.1 [b] | 18.6 ± 0.2 [a] | 17.9 ± 0.5 [b] | 17.7 ± 0.3 [a] | 16.6 ± 0.1 [b] |
| | 24 | 20.4 ± 0.6 [a] | 19.5 ± 0.3 [a] | 19.4 ± 0.6 [a] | 19.1 ± 0.3 [a] | 18.0 ± 0.2 [a] | 17.9 ± 0.6 [a] |
| VOO | 0 | 11.4 ± 0.1 [e] | 11.4 ± 0.1 [e] | 11.4 ± 0.1 [d] | 11.3 ± 0.1 [e] | 11.4 ± 0.1 [e] | 11.5 ± 0.1 [e] |
| | 6 | 13.5 ± 0.1 [d] | 12.2 ± 0.2 [d] | 15.5 ± 0.1 [c] | 13.5 ± 0.3 [d] | 12.4 ± 0.2 [d] | 12.3 ± 0.2 [d] |
| | 12 | 15.9 ± 0.1 [c] | 13.3 ± 0.3 [c] | 16.9 ± 0.2 [b] | 15.4 ± 0.6 [c] | 14.6 ± 0.2 [c] | 13.3 ± 0.1 [c] |
| | 18 | 19.7 ± 0.2 [b] | 15.4 ± 0.2 [b] | 18.7 ± 0.2 [a] | 16.7 ± 0.3 [b] | 16.1 ± 0.1 [b] | 14.1 ± 0.2 [b] |
| | 24 | 21.7 ± 0.5 [a] | 18.6 ± 0.4 [a] | 19.5 ± 0.5 [a] | 19.1 ± 0.2 [a] | 18.3 ± 0.4 [a] | 16.2 ± 0.5 [a] |
| ROO | 0 | 11.7 ± 0.1 [e] | 11.6 ± 0.2 [c] | 11.8 ± 0.1 [d] | 11.6 ± 0.2 [c] | 11.4 ± 0.1 [d] | 11.4 ± 0.1 [c] |
| | 6 | 12.6 ± 0.1 [d] | 12.1 ± 0.1 [b] | 12.3 ± 0.3 [c] | 12.3 ± 0.1 [b] | 12.3 ± 0.2 [c] | 12.2 ± 0.1 [b] |
| | 12 | 23.4 ± 0.2 [c] | 13.5 ± 0.5 [a] | 13.8 ± 0.2 [b] | 12.7 ± 0.6 [b] | 12.5 ± 0.1 [c] | 12.4 ± 0.2 [b] |
| | 18 | 14.7 ± 0.1 [b] | 13.8 ± 0.3 [a] | 14.6 ± 0.1 [a] | 14.3 ± 0.1 [a] | 13.6 ± 0.1 [b] | 12.5 ± 0.6 [b] |
| | 24 | 15.8 ± 0.2 [a] | 14.2 ± 0.7 [a] | 14.8 ± 0.2 [a] | 14.9 ± 0.4 [a] | 14.8 ± 0.2 [a] | 13.4 ± 0.4 [a] |

Results are presented as mean ± standard deviation. BHA—Butylated Hydroxyanisole; BHT—Butylated Hydroxytoluene; EVOO—Extra-virgin Olive Oil; mEq—milliequivalents; MPs—Microparticles; OMP—Olive Mill Pomace; ROO—Blend of virgin olive oil and refined olive oil; VOO—Virgin Olive Oil. Means with the same letter, in the same column, did not differ significantly ($p \leq 0.05$), according to the Tukey test.

At the beginning of the study, both simple (blank), BHA-rich, BHT-rich, BHA + BHT-rich, extract-rich, and MPs-loaded extra-virgin olive oil, virgin olive oil, and blended olive oil presented similar results to PVs (ranged between 11 and 12 mEq/kg), which were within the legal limits: below 20 mEq/kg in the case of EVOOs and VOOs and 15 mEq/kg in the case of ROOs (European Commission, 2019). Similar PVs were found at the beginning of the study by Casal, Malheiro, Sendas, Oliveira and Pereira [34], when the authors compared the oxidative stability of EVOOs, VOOs, and ROOs under deep-frying conditions. The similarity was verified using ANOVA. Along with the study, the PVs of all the oils (EVOO, VOO, and ROO) with or without additives (e.g., BHA, BHT, BHA + BHT, OMP extract, or OMP-loaded ethylcellulose microparticles) increased sharply, except in the case of ROO. In the case of ROO, a decrease in the PV was observed from day 12 to day 18. The decrease in the PV can be attributed to the appearance of hydroperoxides susceptible to decomposition, leading to the formation of carbonyl compounds [41]. Therefore, it can be concluded that the blend of virgin olive oil with refined olive oil presents poor oxidative stability. Similar results were obtained by Yang, Song, Sui, Qi, Wang, Li and Jiang [27], during the analysis of the substitution of synthetic antioxidants by rosemary extract to improve the

oxidative stability of soybean oil, cottonseed oil, and rice bran oil. The authors verified a decrease in the PV of blank cottonseed oil associated with the formation of transient chemical compounds, namely hydroperoxides. Analyses were performed in triplicate for each flask, corresponding to total number of PV analyses of 9.

On day 24, the blank of all oils presented PVs (20.4 ± 0.6 mEq/kg in the case of EVOO; 21.7 ± 0.5 mEq/kg in the case of VOO and 15.8 ± 0.2 mEq/kg in the case of ROO) higher than the maximum legal limit (20 mEq/kg in the case of EVOOs and VOOs and 15 mEq/kg in the case of ROOs). The incorporation of synthetic antioxidants (BHA, BHT, or a blend of BHA and BHT) or natural antioxidants (OMP extract or OMP-loaded ethylcellulose microparticles) prevented oils from excessive oxidation; the embedment favored the oxidative stability of olive oils. On day 24, the blank EVOO, VOO, and ROO exhibited ($p < 0.05$) higher PVs compared to EVOO, VOO and ROO enriched with BHA, BHT, a blend of BHA and BHT, extract, or microparticles. Among the OOs enriched with synthetic antioxidants, the PVs were not significantly different ($p > 0.05$). The PVs of oils enriched with extract and microparticles at the end of the study (24th day) were statistically significant ($p < 0.05$). Oils incorporating OMP-loaded ethylcellulose microparticles demonstrated significantly lower PVs than the obtained for enriched oils patterns (olive oils enriched with synthetic antioxidants and OMP extract). The OMP-loaded ethylcellulose microparticles were more efficient in the retardation of peroxidation phenomena. The observed PVs of oils enriched with microparticles were below the maximum legal limit for PVs of olive oils. Olive oils enriched with OMP-extract-loaded ethylcellulose microparticles exhibited a great potential to retard the formation of hydroperoxides.

### 3.3. Changes on the p-Anisidine Values (p-AVs)

The *p*-AV measures the α- and β-alkenals as secondary oxidation products [31]. *p*-anisidine is a compound that strongly reacts with aldehydes forming products that absorb at a wavelength of 350 nm. It corresponds to the absorbance of a solution resulting from the reaction of the fat or oil with *p*-anisidine in an isooctane solution. The products formed by reaction with unsaturated aldehydes, namely 2-alkenals, absorb intensely at 350 nm. Therefore, the *p*-AV tests are exceptionally sensitive to these oxidation products. The *p*-AV is a measure of the presence of secondary oxidation products, namely ketones, aldehydes, and carboxyl acids [29,42]. Results regarding the *p*-AV levels of EVOO, VOO, and ROO during the 24-day storage are shown in Figure 1.

After 24 days from the beginning of the study, each oil—blank and enriched—had a higher *p*-AV than their initial values on day 0. All the olive oil samples (EVOO, VOO, and ROO) exhibited identical evolution on the *p*-AVs during the study, independently of the initial PVs, *p*-AVs, and their variations. The formation of stable secondary oxidation species occurred at similar rates in EVOO, VOO, and ROO samples. During the study of olive oils stability under deep-frying conditions, the authors concluded that in extra-virgin olive oils, virgin olive oils, and blends of refined olive oils and virgin oils, the *p*-AVs evolution was independent of the initial PVs and *p*-AVs as well as their variations during the study. The *p*-AVs of blank oils and enriched oils were statistically significant ($p < 0.05$). No differences ($p > 0.05$) were found between the *p*-AVs of oils enriched with synthetic antioxidants (BHA and BHT), considered solely or in a blend. Notwithstanding, the embedment of OMP-extract microparticles led to the obtainment of significantly ($p < 0.05$) lower *p*-AVs, which was related to a lower content of secondary oxidation products in EVOO, VOO, and ROO selected in the present study. In the present study, for all oils, it was observed that *p*-AVs and *p*-AV change rates were higher than the reported values by Asensio, Nepote and Grosso [42]; the authors were evaluating the effect of adding oregano essential oil to extra-virgin olive oil during long-term storage conditions (maximum *p*-AV observed was about 9.3, after 28 days of storage). Nevertheless, the obtained *p*-AVs in the present study were lower than the ones presented by Casal, Malheiro, Sendas, Oliveira and Pereira [34] (maximum *p*-AV of 66 in an EVOO, after 15 h of frying). This current study supports the thesis that encapsulated antioxidants are sustainably released from microparticles

embedded in olive oils, effectively reducing the presence of stable secondary oxidation products through antioxidants acting as hydrogen-donating substituents.

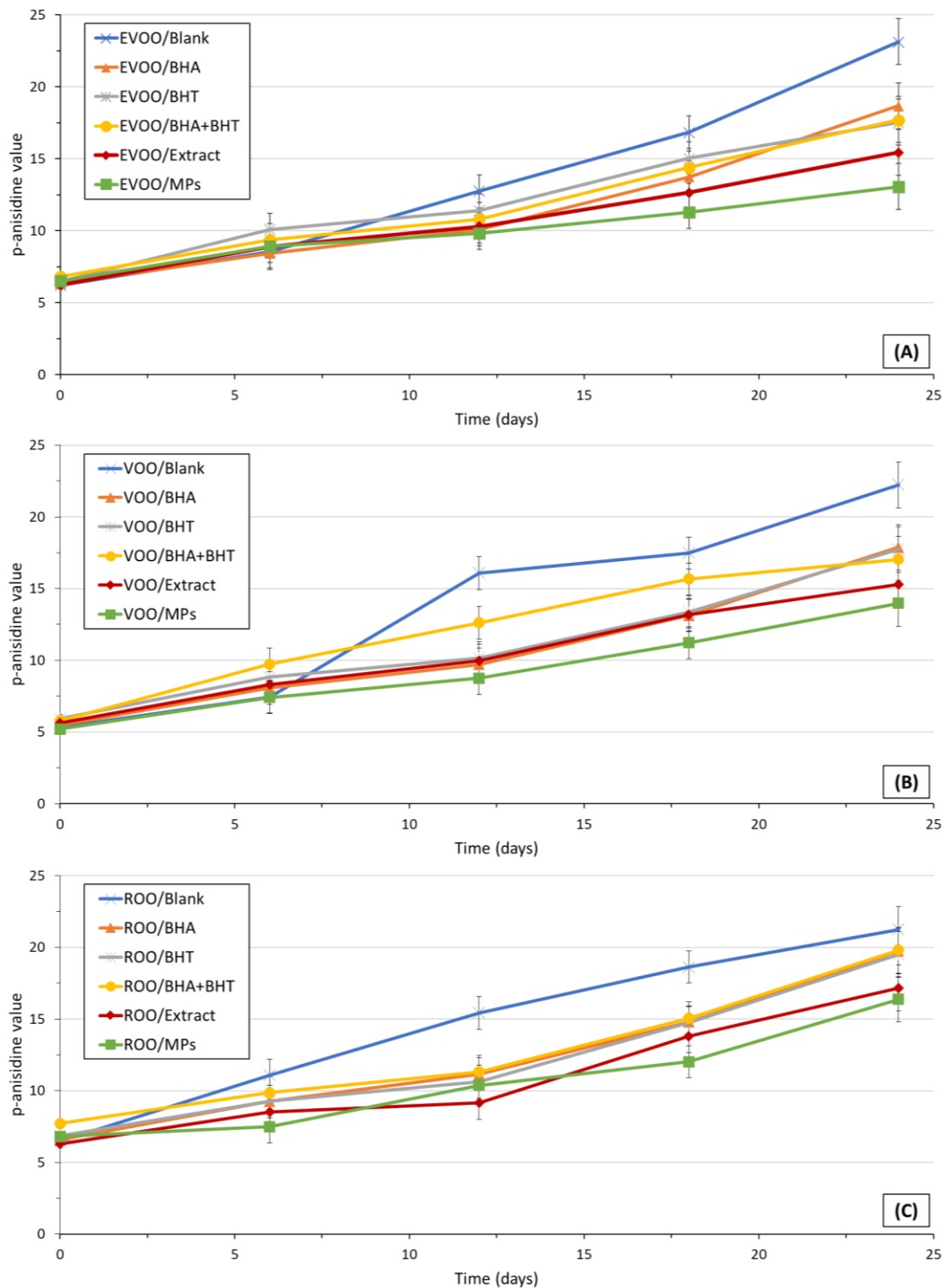

**Figure 1.** Changes in the *p*-anisidine value of plain ( ) and incorporating BHA ( ), BHT ( ), BHA + BHT ( ), extract ( ) and microparticles ( ) extra-virgin olive oil (**A**), virgin olive oil (**B**), and a blend of refined and virgin oil (**C**) during storage (BHA—Butylated Hydroxyanisole; BHT—Butylated Hydroxytoluene; EVOO—Extra-Virgin Olive Oil; MPs—Microparticles; ROO—Blend of refined and virgin olive oils; VOO—Virgin Olive Oil).

### 3.4. Changes in the Total Oxidation (TOTOX) Values

The TOTOX value is an indicator of the overall oxidation status of an oil [33]. It is generally acknowledged that the empirical maximum acceptable level of TOTOX value is 30 (Sun-Waterhouse et al., 2011). Results regarding the obtained TOTOX values based on PVs and *p*-AVs are presented in Table 4.

**Table 4.** Total oxidation values of olive oils during storage.

| Olive Oil Sample | Storage Time (Days) | TOTOX Value [a] | | | | | |
|---|---|---|---|---|---|---|---|
| | | **Blank** | **BHA** | **BHT** | **BHA + BHT** | **OMP Extract** | **MPs** |
| EVOO | 0 | 17.7 ± 0.4 [e] | 17.6 ± 0.4 [e] | 18.0 ± 0.6 [e] | 18.0 ± 0.6 [e] | 17.8 ± 0.2 [e] | 18.0 ± 0.5 [d] |
| | 6 | 24.0 ± 0.3 [d] | 21.8 ± 0.4 [d] | 24.6 ± 0.4 [d] | 23.3 ± 0.3 [d] | 22.0 ± 0.2 [d] | 21.0 ± 1.2 [c] |
| | 12 | 30.4 ± 0.4 [c] | 25.9 ± 0.4 [c] | 27.1 ± 0.3 [c] | 26.2 ± 0.8 [c] | 25.2 ± 0.3 [c] | 25.6 ± 1.4 [b] |
| | 18 | **35.4 ± 0.2** [b] | **31.5 ± 0.3** [b] | **33.7 ± 0.4** [b] | **32.3 ± 0.8** [b] | 30.3 ± 0.5 [b] | 27.9 ± 1.2 [b] |
| | 24 | **43.5 ± 0.9** [a] | **38.2 ± 0.6** [a] | **36.9 ± 1.2** [a] | **36.8 ± 0.7** [a] | **33.5 ± 0.4** [a] | 31.0 ± 1.2 [a] |
| VOO | 0 | 16.7 ± 0.2 [e] | 16.9 ± 0.2 [e] | 17.3 ± 0.2 [f] | 17.1 ± 0.4 [e] | 17.0 ± 0.2 [e] | 16.7 ± 0.2 [e] |
| | 6 | 20.9 ± 0.3 [d] | 20.3 ± 0.4 [d] | 24.3 ± 0.2 [d] | 23.2 ± 0.4 [d] | 20.7 ± 0.3 [d] | 19.7 ± 0.3 [d] |
| | 12 | **32.0 ± 0.4** [c] | 23.0 ± 0.5 [c] | 27.1 ± 0.4 [c] | 28.0 ± 1.5 [c] | 24.6 ± 0.3 [c] | 22.1 ± 0.2 [c] |
| | 18 | **37.2 ± 0.4** [b] | 28.5 ± 0.4 [b] | **32.1 ± 0.4** [b] | **32.4 ± 0.6** [b] | 29.3 ± 0.3 [b] | 25.3 ± 0.2 [b] |
| | 24 | **43.9 ± 0.9** [a] | **36.5 ± 0.6** [a] | **37.2 ± 0.1** [a] | **36.1 ± 0.8** [a] | **33.6 ± 0.5** [a] | 30.2 ± 0.4 [a] |
| ROO | 0 | 18.2 ± 0.3 [d] | 18.2 ± 0.4 [e] | 18.7 ± 0.2 [e] | 19.3 ± 0.3 [e] | 17.7 ± 0.2 [e] | 18.2 ± 0.1 [e] |
| | 6 | 23.7 ± 0.2 [c] | 21.4 ± 0.2 [d] | 21.6 ± 0.5 [d] | 22.2 ± 0.2 [d] | 20.8 ± 0.4 [d] | 19.7 ± 0.2 [d] |
| | 12 | **38.8 ± 0.6** [a] | 24.7 ± 0.7 [c] | 24.4 ± 0.4 [c] | 24.0 ± 1.0 [c] | 21.6 ± 0.2 [c] | 22.8 ± 0.2 [c] |
| | 18 | **33.3 ± 0.4** [b] | 28.6 ± 0.5 [b] | 29.3 ± 0.2 [b] | 29.4 ± 0.2 [b] | 27.4 ± 0.2 [b] | 24.5 ± 0.7 [b] |
| | 24 | **37.0 ± 0.9** [a] | 33.9 ± 1.4 [a] | 34.3 ± 0.4 [a] | 34.7 ± 0.4 [a] | 31.9 ± 0.4 [a] | 29.8 ± 0.6 [a] |

Results are presented as mean ± standard deviation. Data in bold and marked in a grey background exceeded the maximum acceptable level of the TOTOX value. BHA—Butylated Hydroxyanisole; BHT—Butylated Hydroxytoluene; EVOO—Extra-virgin Olive Oil; MPs—Microparticles; OMP—Olive Mill Pomace; ROO—Blend of virgin olive oil and refined olive oil; TOTOX—Total Oxidation; VOO—Virgin Olive Oil. Means with the same letter, in the same column, did not differ significantly ($p \leq 0.05$), according to the Tukey test.

Generically, the TOTOX values increased with the increase in storage time of all olive oil samples. At day 24 and in the case of EVOO, the TOTOX values decreased in the order of blank > BHA > BHT > BHA + BHT > OMP extract > microparticles. In the case of both VOO and ROO, on day 24, it was observed that the TOTOX values decreased in the order of blank > BHA > BHT > BHA + BHT > OMP extract > microparticles. In all oil samples, the lowest TOTOX values were observed in the case of the enrichment with microparticles, followed by oils incorporation with OMP extracts. The TOTOX values exceeding the maximum acceptable level (30) are marked in the grey background in Table 4. At the end of the storage period, it was observed that the quality of oils embedding encapsulated antioxidants was acceptable. The quality of oils embedding OMP extract was guaranteed until day 18 of the study. Nevertheless, the quality of plain oils and oils fortified with BHA, BHT, or with a blend of BHA and BHT was not acceptable from day 18 onwards (TOTOX values > 30). The obtained results in the current study are in agreement with the ones obtained by Sun-Waterhouse, Zhou, Miskelly, Wibisono and Wadhwa [29], during the study of the stability of an extra-virgin oil fortified with encapsulated caffeic acid. The authors verified an increase in oil quality (reduction in the TOTOX values) from the blank to enriched oil with encapsulated caffeic acid. The low TOTOX values are correlated with the high quality of oils. In the present study, it was verified that the quality of extra-virgin olive oil, virgin olive oil, and a blend of virgin oil and refined olive oil was improved when fortified with encapsulated antioxidants obtained from olive mill pomace.

### 3.5. Changes in the Free Fatty Acids (FFAs) Content

The monitorization of changes in free fatty acids (FFAs) content is considered to be relevant to measure rancidity in foods, especially in oils and fats. The FFAs are obtained through

the hydrolysis of triglycerides; the reaction of oils with moisture prompts the formation of these compounds [43]. Results regarding the acidity values are presented in Figure 2.

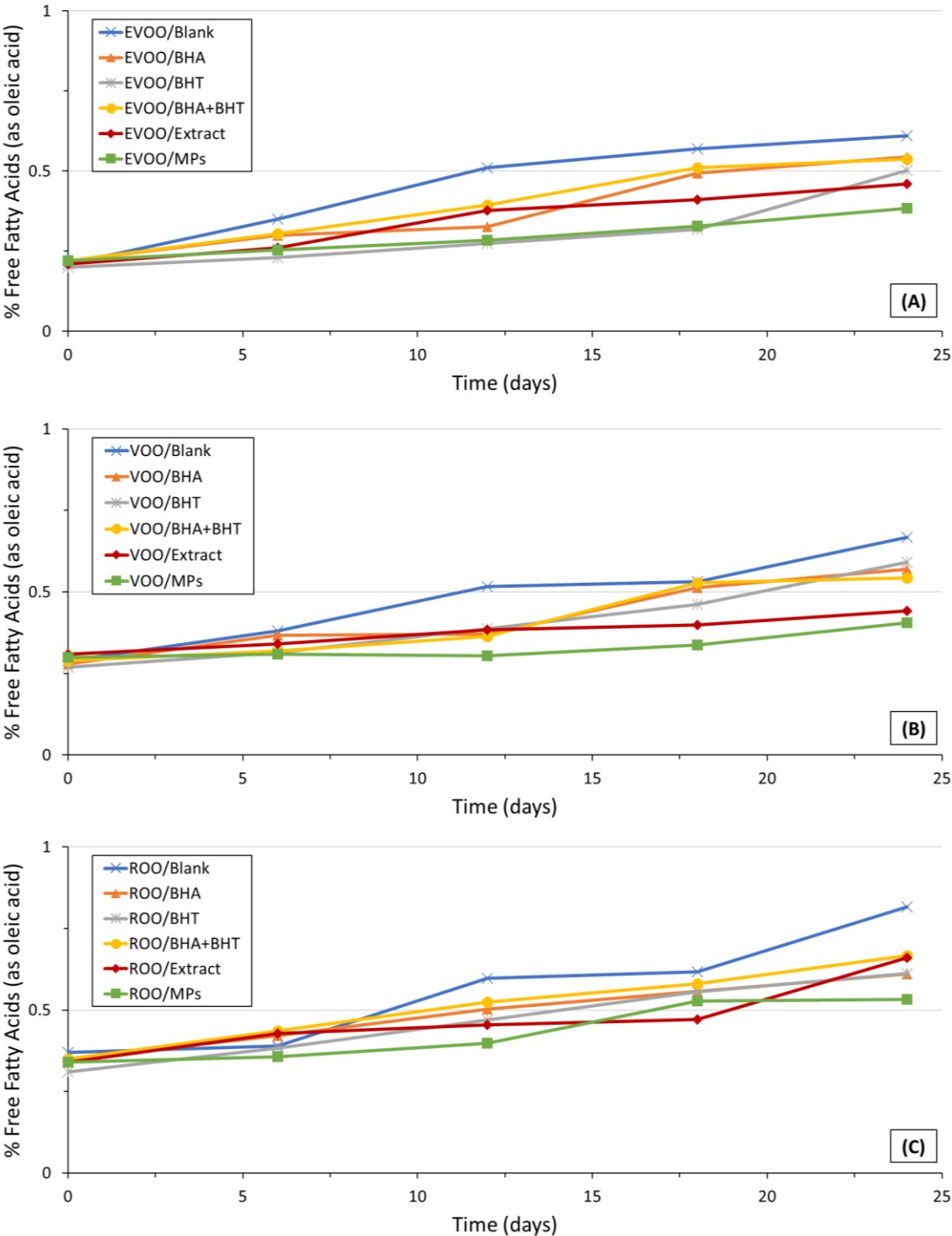

**Figure 2.** Changes in free acidity values (% free fatty acids as oleic acid) of plain ( —✕— ) and incorporating BHA ( —▲— ), BHT ( —✳— ), BHA + BHT ( —●— ), extract ( —◆— ) and microparticles ( —■— ) extra-virgin olive oil (**A**), virgin olive oil (**B**), and a blend of refined and virgin oil (**C**) during storage. (BHA—Butylated Hydroxyanisole; BHT—Butylated Hydroxytoluene; EVOO—Extra-Virgin Olive Oil; MPs—Microparticles; ROO—blend of refined and virgin olive oils; VOO—Virgin Olive Oil).

For all olive oil samples, plain and enriched, an increase in the FFA content was observed with the increase in the storage period. Nevertheless, in the present study, a regular pattern of the increase was not observed. Similar conclusions were drawn by Iqbal and Bhanger [44]: similarly, the authors verified during their study regarding the stabilization of sunflower oil using an extract of garlic acid, that the FFA content increased with the increase in the storage period, not identifying an increasing pattern. During the current study, the FFA content was below 0.62% in the case of EVOO, 0.67% in the case of VOO, and below 0.83% in the case of ROO. The obtained acidity values were in agreement with the respective labeling category, the FFA (%) should be lower than 0.80%, 2.00%, and 1.00%, in the case of extra-virgin olive oils, virgin olive oils, and blends of refined and virgin olive oils, respectively [35]. In all oil samples, a higher ($p < 0.05$) acidity of the plain oils was observed compared to the enriched ones. In both EVOO, VOO, and ROO, the acidity of oils enriched with microparticles was significantly lower than those enriched with synthetic antioxidants (BHA, BHT, or a blend of these antioxidants). Nevertheless, the acidity of oils enriched with OMP-loaded ethylcellulose microparticles and OMP extract only were similar ($p > 0.05$) for all time points during the storage study. Therefore, it can be concluded that the incorporation of olive mill pomace extracts plain or formulated in microparticles, favored the hydrolytic stability of the oils, namely through the control of the acidity of the oils (lower values of oils acidity enriched with extract and microparticles rather than plain oils and oils incorporating synthetic antioxidants). Similarly, the authors Sun-Waterhouse, Zhou, Miskelly, Wibisono and Wadhwa [29], verified that encapsulated and non-encapsulated caffeic acid was effective in slowing down FFA production in an extra-virgin oil. The ROOs presented higher acidities at all time points rather than EVOO and VOO. This observation can be explained considering that (i) the ROO is a blend of refined and virgin olive oils and (ii) olive oil refining prompts the removal of free fatty acids by saponification. In this context, only trace amounts of FFA are expected in refined olive oils. However, as slighter higher acidity values were observed for the ROOs, it can be hypothesized that the virgin olive oil present in the blend presented higher acidity than the VOO considered solely in the study. The results of this study indicate the incorporation of olive mill pomace extract and olive mill pomace extract encapsulated into microparticles in olive oils (extra-virgin olive oil, virgin olive oil, and a blend of refined and virgin olive oil) effectively slowed down the hydrolysis, ensuring the acidity values were below the rejection points even during accelerated storage conditions. Moreover, in this study, it is proven that natural-occurring antioxidants presented in an extract or formulated in microparticles are more effective in retarding the free fatty acids formation than synthetic antioxidants. To retard hydrolysis processes in oil samples, it is recommended to add natural-occurring antioxidants rather than the synthetic ones (e.g., OMP or encapsulated OMP extract over BHA and BHT).

### 3.6. Changes on the $K_{232}$ and $K_{270}$ Extinction Coefficients

The ultraviolet spectrophotometric analysis through the specific extinction coefficients provides some useful insights into the olive oil oxidation status. The $K_{232}$ extinction coefficient is related to the formation of conjugated dienes of polyunsaturated fatty acids (PUFA), whereas the $K_{270}$ extinction coefficient value indicates the presence of both primary and secondary oxidation products, namely carbonyl compounds and conjugated [34]. The maximum admissible of the $K_{232}$ extinction coefficient is 2.5 in the case of EVOOs, and 2.6 in the case of VOOs. Nevertheless, in the case of blends of virgin and refined olive oils, the legal limit of $K_{232}$ is not defined. The legal limit of $K_{270}$ is 0.22, 0.25, and 1.15 in the case of EVOOs, VOOs, and ROOs, respectively [35]. All olive oil samples presented $K_{232}$ extinction coefficients within the legislation limits at all-time points during the study (Figure 3).

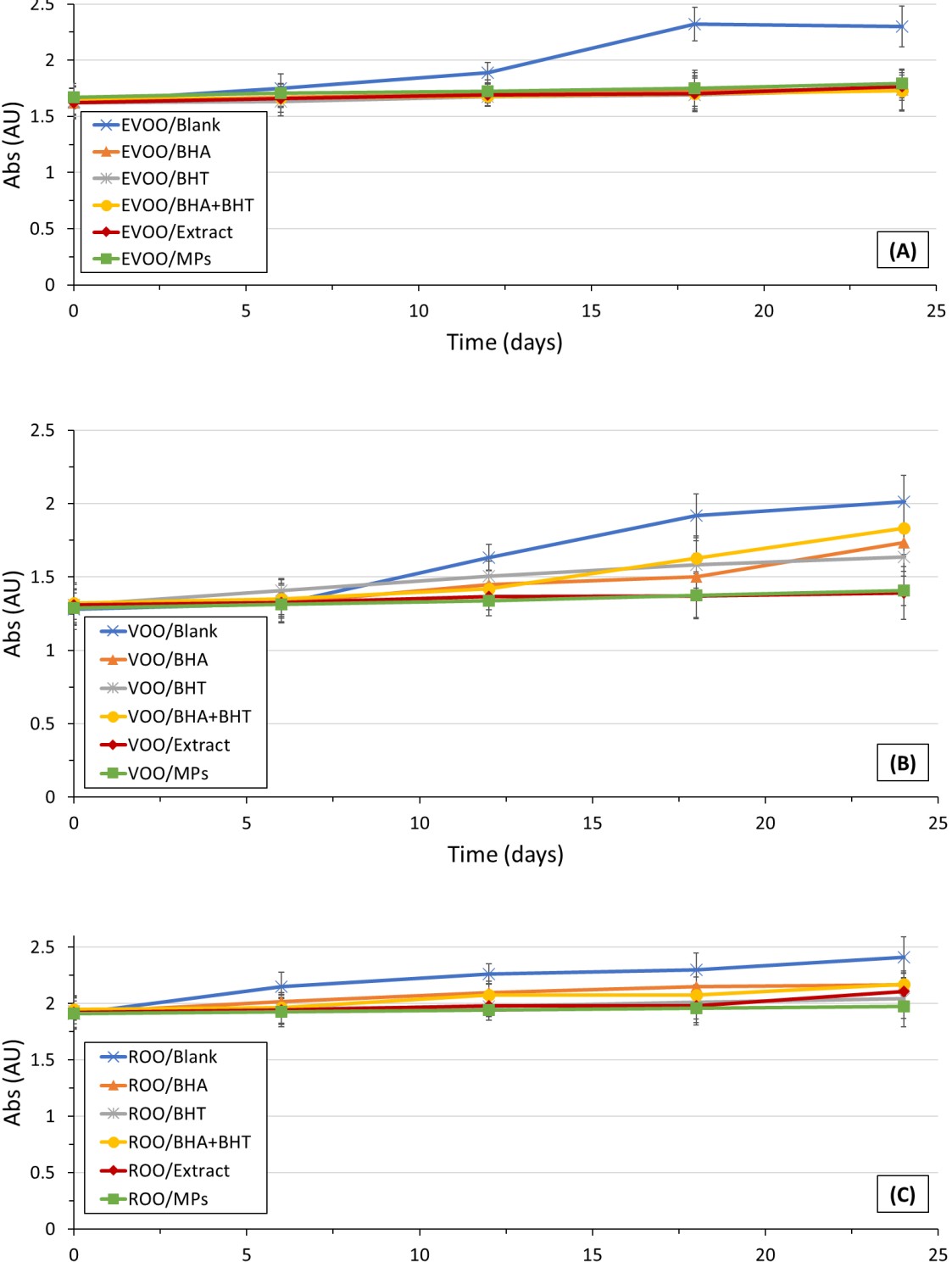

**Figure 3.** Changes on the $K_{232}$ extinction coefficient (Abs; AU) of plain ( ) and incorporating BHA ( ), BHT ( ), BHA + BHT ( ), extract ( ) and microparticles ( ) extra-virgin olive oil (**A**), virgin olive oil (**B**), and a blend of refined and virgin oil (**C**) during storage (BHA—Butylated Hydroxyanisole; BHT—Butylated Hydroxytoluene; EVOO—Extra-Virgin Olive Oil; MPs—Microparticles; ROO—Blend of refined and virgin olive oils; VOO—Virgin Olive Oil).

In the case of plain EVOO and VOO, significant differences were observed ($p < 0.05$) in the $K_{232}$ values compared to natural and synthetic antioxidant patterns. Differences were not observed between oil samples enriched with BHA, BHT, a blend of BHA and BHT, extract, and OMP-loaded ethylcellulose microparticles in all oil samples (EVOO, VOO, and ROO). A plateau in the $K_{232}$ values in the case of fortified EVOO during storage was observed. Therefore, it can be concluded that the incorporation of both natural and synthetic antioxidants favored olive oil stability during storage. Probably, only a minor amount of conjugated dienes of PUFA were produced during the 24-day storage under accelerated conditions. The evolution of the $K_{270}$ extinction coefficients during storage is presented in Figure 4.

In the case of the EVOO and the VOO, differences ($p > 0.05$) were not observed among oil samples (fortified and non-fortified). In the case of the EVOO, the $K_{270}$ values were kept below the legal limit (<0.22; maximum observed $K_{270}$ value of $0.22 \pm 0.42$ in the case EVOO fortified with a blend of BHA and BHT), except in the case of the plain EVOO at day 24 where a $K_{270}$ value slightly above the legal limit ($0.61 \pm 0.23$) was observed. In the case of the EVOO, the $K_{270}$ extinction coefficient was independent of the incorporation of natural or even synthetic antioxidants. The fortification of extra-virgin olive oil samples with antioxidants (natural; natural embedded in microparticles; synthetic) did not impact the $K_{270}$ extinction coefficient. Dissimilarly, the $K_{270}$ extinction coefficients of the plain ROO and the ROO fortified with OMP-loaded ethylcellulose microparticles were significantly different. On day 24, the $K_{270}$ of ROO incorporating OMP-loaded ethylcellulose microparticles ($K_{270}$ observed value of $0.24 \pm 0.07$) was lower ($p < 0.05$) than the plain ROO ($K_{270}$ observed value of $0.66 \pm 0.09$), the ROO fortified with synthetic antioxidants ($K_{270}$ observed value of $0.42 \pm 0.08$ in the case of fortification with BHA; $K_{270}$ of $0.50 \pm 0.02$ in the case of BHT and $0.44 \pm 0.01$ in the case of the fortification with a blend of BHA and BHT) and the ROO incorporating an extract of OMP ($K_{270}$ observed value of $0.47 \pm 0.03$). In all ROO oil samples, the $K_{270}$ extinction coefficients were kept below the legal limit (1.15) throughout the 24-day study. The present study results are partially in accordance with the ones obtained by Casal, Malheiro, Sendas, Oliveira and Pereira [34], during their study of olive oil stability under deep-frying conditions. The authors also observed an increase in the $K_{232}$ parameter with time, not verifying, similarly to the present study, whether $K_{232}$ values exceed the legal limit. Similarly, the authors did not observe clear differences between oil samples.

### 3.7. Changes in the Total Antioxidant Activity (TAA) and the Total Phenolic Content (TPC)

The evaluation of the total antioxidant activity (TAA) of oils is commonly employed to determine the stability of oils [27,45]. Results are presented in Figure 5 A–C.

From the ANOVA analysis, no significant differences ($p > 0.05$) were found between the TAA values at days 6, 12, and 18 for all oil samples. A similar observation was drawn by Yang, Song, Sui, Qi, Wang, Li and Jiang [27] during their study of the stability of a vegetable oil enriched with rosemary extract during storage (Schaal oven test; 24-day study). Similarly, the authors did not verify differences in the TAA of oils (plain and fortified) from day 6 to day 18. The highest TAA, at day 0, was observed in the case of oils enriched with OMP extract (TAA of EVOO enriched with extract of $21.1 \pm 0.2$ mg TE/100 mL of oil; TAA of VOO fortified with OMP extract of $18.3 \pm 0.1$ mg TE/100 mL of oil and TAA of ROO incorporating OMP extract of $21.2 \pm 0.9$ mg TE/100 mL of oil). At the beginning of the study, oils fortified with OMP extract demonstrated higher antioxidant activity than blank oils, oils fortified with synthetic antioxidants (BHA, BHT, and a blend of BHA and BHT), and oils enriched with OMP extract-loaded ethylcellulose microparticles. The TAA of oils incorporating OMP extract-loaded ethylcellulose microparticles was similar ($p < 0.05$) to plain oils. This was an expected result, as the coating material of microparticles—ethylcellulose—does not exhibit antioxidant activity, and the majority of antioxidants were efficiently embedded inside of microparticles and not adsorbed on the surface of microparticles, which explains the TAA values of oils incorporating microparticles similar to the TAA values of plain oils. During the 24-day study, the TAA of plain oils, oils fortified with synthetic antioxidants (BHA,

BHT, and a blend of BHA and BHT), and oils enriched with OMP extract significantly decreased ($p < 0.05$), except in the case of oils incorporating microparticles. In the case of olive oils fortified with OMP-loaded ethylcellulose microparticles, the TAA was kept barely constant during the 24-day storage study (e.g., TAA of EVOO/MPs at day 0 of $12.2 \pm 0.1$ mg TE/100 mL oil; TAA of EVOO/MPs at day 24 of $11.3 \pm 0.2$ mg TE/100 mL oil). Therefore, it can be concluded that olive mill pomace-loaded ethylcellulose microparticles have proven to be an efficient delivery system of antioxidants in olive oils to increase the oil's oxidative stability during storage. During the current study, differences between the TAA of oils fortified with synthetic antioxidants (BHA, BHT, or a blend of them) were not observed. The higher TAA of oils with OMP extract compared with oils enriched with synthetic antioxidants can be explained by considering that hydroxytyrosol is the major bioactive compound in the OMP extract exhibiting outstanding antioxidant activity [46–48]. Hydroxytyrosol presents two *o*-phenolic hydroxyl groups bonded to the benzene ring, while BHA and BHT present only one hydroxyl group attached to the benzene ring. Even though the TAA is affected by the relative position of *o*-phenolic hydroxyl groups and the presence of specific groups in the molecular structure, it is generally recognized that the TAA is mainly affected by the number of *o*-phenolic hydroxyl groups in the molecular structure [27]. Therefore, the increase in the number of *o*-phenolic hydroxyl groups from BHA and BHT to hydroxytyrosol led to an increase in the TAA of oils incorporating OMP extract. Olive oils are receiving particular attention compared to other vegetable oils due to their high content of phenolic compounds, which bestow olive oil's antioxidant and even antimicrobial properties and their noteworthy role in the maintenance of olive oil's oxidative status [49,50]. The TPC of olive oils was monitored through 24 days of accelerated storage conditions. Results are presented in Figure 5D–F. From the ANOVA, data from days 8, 12, and 18 are not presented as no significant differences ($p > 0.05$) were found between them for all oil samples. In comparison, blank EVOO presented higher TPC ($198.3 \pm 12.1$ mg$_{GAE}$/kg oil) than ROO ($124.5 \pm 6.8$ mg$_{GAE}$/kg oil) and VOO ($87.2 \pm 10.5$ mg$_{GAE}$/kg oil). These results are in agreement with the ones obtained by Casal et al. (2010). Similarly, the authors observed higher TPC values on EVOO, followed by ROO, detecting the lowest TPC values in VOOs. At the beginning of the study (day 0), all oil samples (EVOO, VOO, and ROO) incorporating OMP extract exhibited significantly ($p < 0.05$) higher TPC than their blank, or samples enriched with synthetic (BHA, BHT and a blend of BHA and BHT) and microparticle oils patterns. The total phenolic content of olive oil samples significantly decreased ($p < 0.05$) during storage, except for the case of oils enriched with OMP-extract loaded ethylcellulose microparticles. The TPC of oil samples with extract and microparticles on day 24 was similar ($p > 0.05$). The observed decrease in the TPC probably was due to the oxidation and decomposition of phenolic compounds present in the oil matrices, which experience both qualitative and quantitative physicochemical modifications during storage [51]. Moreover, in oils, phenolic compounds act as antioxidants by donating H-atom(s) to free radicals, contributing to the decrease in the TPC [52]. However, in the case of oils with microparticles, the phenolic compounds are controlled dripping. This prompts a balance between the decomposition and the release of phenolic compounds, which allows for maintaining the oxidative stability of olive oils. These results support the thesis that olive mill pomace-loaded ethylcellulose microparticles were efficiently designed to sustain the release of phenolic compounds to control the oxidative status of oil samples. The incorporation of OMP extract-loaded ethylcellulose microparticles improved the oxidative resistance of olive oils.

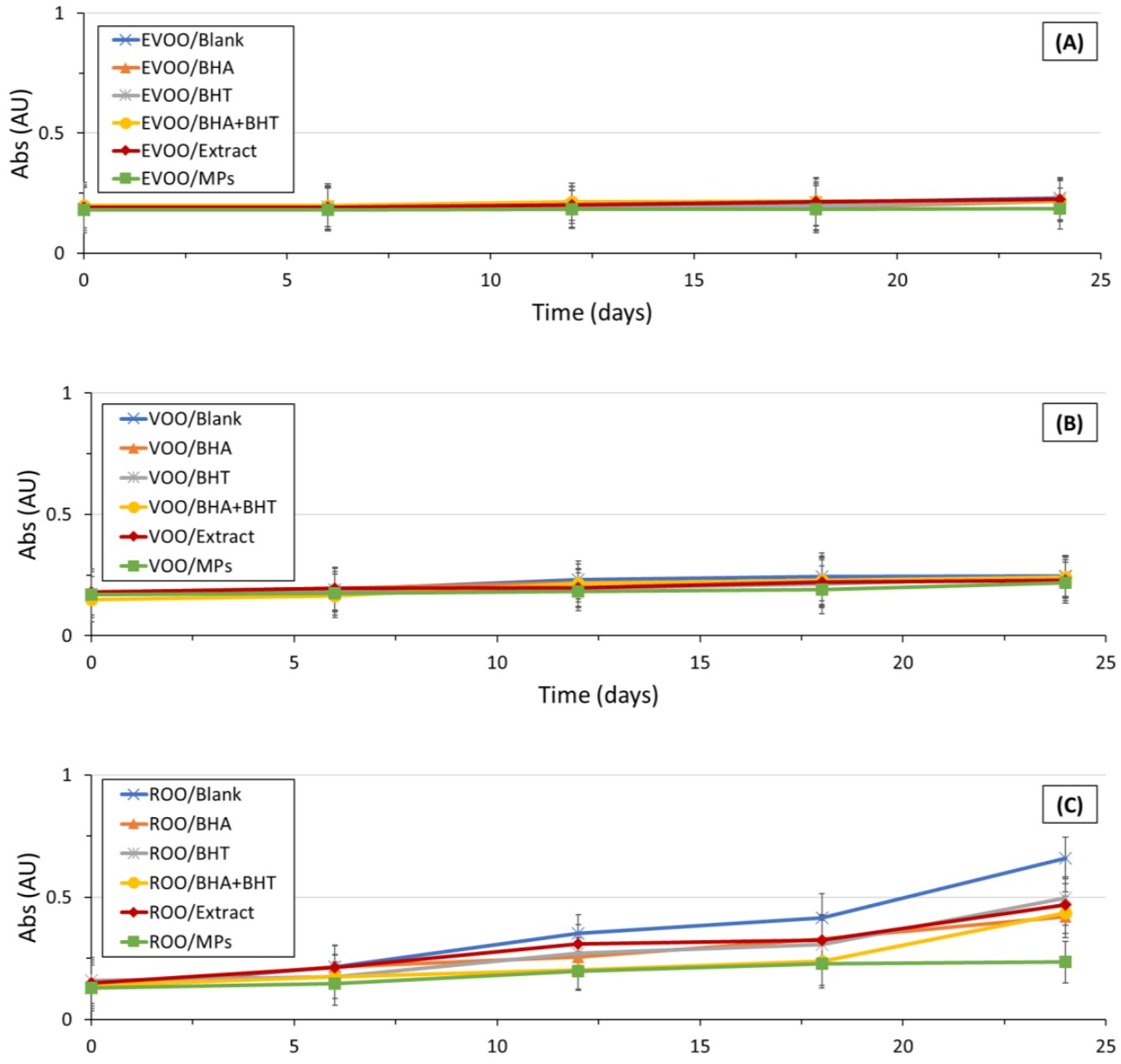

**Figure 4.** Changes on the $K_{270}$ extinction coefficient (Abs; AU) of plain (  ) and incorporating BHA (  ), BHT (  ), BHA + BHT (  ), extract (  ) and microparticles (  ) extra-virgin olive oil (**A**), virgin olive oil (**B**), and a blend of refined and virgin oil (**C**) during storage. (BHA—Butylated Hydroxyanisole; BHT—Butylated Hydroxytoluene; MPs—Microparticles).

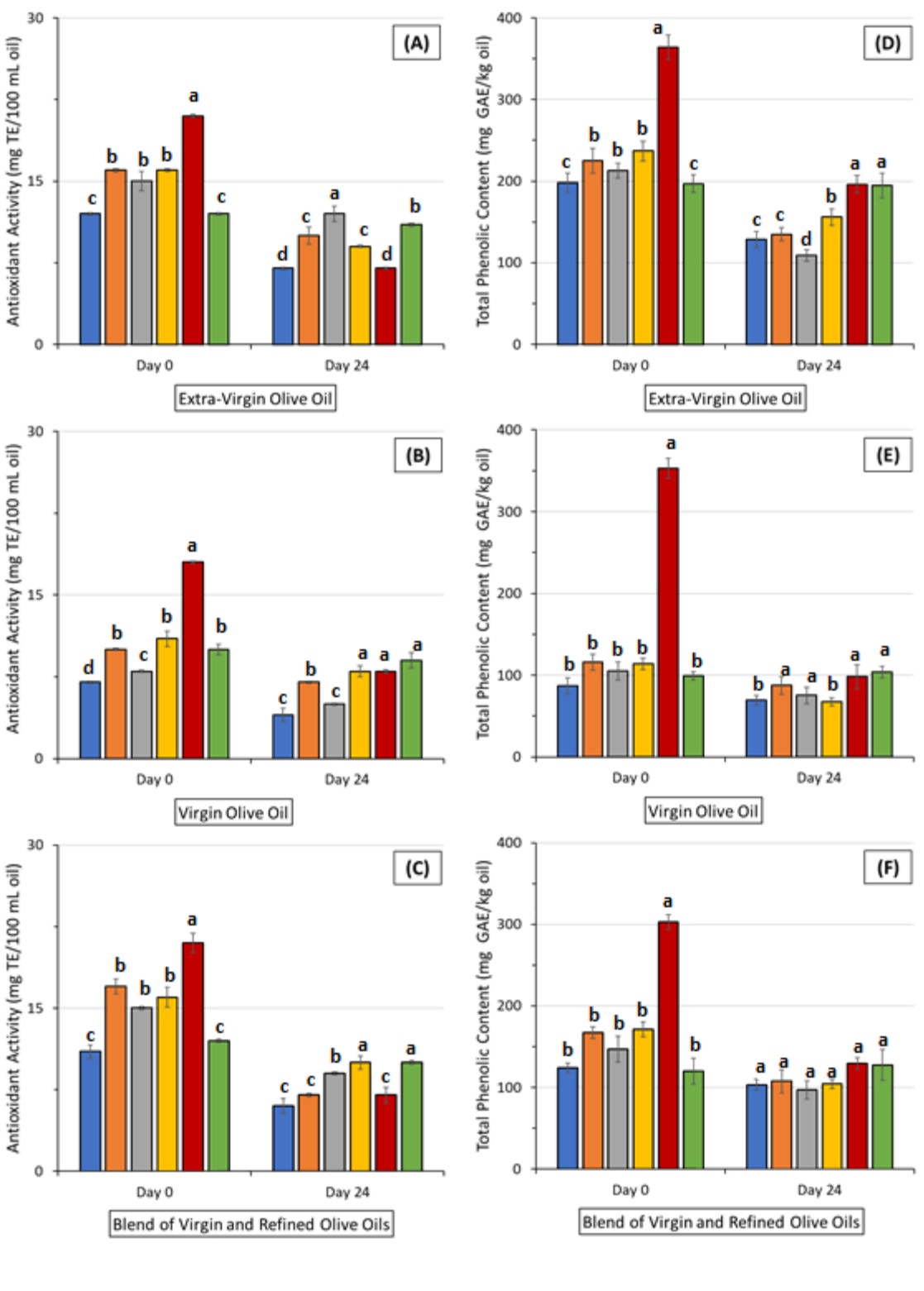

**Figure 5.** Results of changes in the antioxidant capacity measured by the DPPH of an extra-virgin olive oil (**A**), a virgin olive oil (**B**) and a blend of virgin and refined olive oils (**C**) and total phenolic content of an extra-virgin olive oil (**D**), a virgin olive oil (**E**) and a blend of virgin and refined olive oils (**F**) during storage. (BHA—Butylated Hydroxyanisole; BHT—Butylated Hydroxytoluene; GAE—Gallic Acid Equivalents; MPs—Microparticles; TE—Trolox Equivalents). Means with the same letter, in the same column, did not differ significantly ($p \leq 0.05$), according to the Tukey test.

## 4. Conclusions

The bioactive compounds present in olive mill pomace extracts were efficiently encapsulated in ethylcellulose microparticles. The oxidative stability indices, namely the peroxide value, the *p*-anisidine value, the total oxidation value, the free fatty acids content, the $K_{232}$ and $K_{270}$ extinction coefficients, the total antioxidant activity, and the total phenolic content of plain olive oils, olive oils enriched with synthetic antioxidants (butylated hydroxyanisole, butylated hydroxytoluene and a blend of them), olive oils with olive mill pomace extract and oils fortified with olive mill pomace loaded ethylcellulose microparticles, were appraised during accelerated storage conditions. Olive oil samples fortified with microparticles exhibited lower peroxide values than plain and enriched respective patterns. Olive mill pomace-loaded ethylcellulose microparticles were shown to be efficient in the retardation of peroxidation processes. The embedment of olive mill pomace extract-loaded microparticles led to the obtainment of lower *p*-anisidine values; therefore, the rate of hydroperoxide generation was lower compared to plain and enriched (synthetic antioxidants and olive mill pomace extract) olive oils. Therefore, it can be concluded that loaded microparticles efficiently lower the content of secondary oxidation products in extra-virgin olive oils, virgin olive oils, and blends of refined and virgin olive oils. The overall quality of extra-virgin olive oil, virgin olive oil, and a blend of virgin oil and refined olive oil was improved when fortified with encapsulated antioxidants obtained from olive mill pomace proven by the total oxidation values obtained. The incorporation of olive mill pomace extract and olive mill pomace extract encapsulated into microparticles in olive oils effectively slowed down hydrolysis processes, ensuring the acidity values were kept below the rejection points even during accelerated storage conditions. The total antioxidant activity and the total phenolic content of oils incorporating microparticles remained barely constant during storage time. Briefly, the current study brings new insights into the design of natural antioxidants-rich loaded microparticles that can be further incorporated with oils to retard oxidation processes occurring in the olive oil matrices. The design of antioxidants-rich loaded microparticles may efficiently improve the oil's oxidative stability.

**Author Contributions:** Conceptualization, F.P. and L.S.; methodology, F.P. and L.S.; validation, F.P., L.T. and L.S.; formal analysis, F.P., L.T. and L.S.; investigation, F.P.; resources, L.S.; data curation, F.P. and L.T.; writing—original draft preparation, F.P., L.T. and L.S.; writing—review and editing, F.P., L.T. and L.S.; visualization, F.P., L.T. and L.S.; supervision, L.S.; project administration, L.S.; funding acquisition, L.S. All authors have read and agreed to the published version of the manuscript.

**Funding:** This work was financially supported by the grant NORTE-08-5369-FSE-000028, co-financed by the Northern Regional Operational Program (NORTE 2020) through Portugal 2020 and the European Social Fund (ESF). This work was developed under the doc-toral program in Chemical and Biological Engineering (PDEQB), financially supported by the grant NORTE-08-5369-FSE-000028.

**Data Availability Statement:** The data presented in this study are available on request from the corresponding authors.

**Acknowledgments:** This work was financially supported by: UIDB/00511/2020, LA/P/0045/2020 (ALiCE), and UIDP/00511/2020 (LEPABE), funded by national funds through FCT/MCTES (PID-DAC); This work was developed under the doctoral program in Chemical and Biological Engineering (PDEQB) NORTE-08-5369-FSE-000028, co-financed by the Northern Regional Operational Program (NORTE 2020) through Portugal 2020 and the European Social Fund (ESF).

**Conflicts of Interest:** The authors declare no conflict of interest.

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
