# Peer review of "Olive Mill Pomace Extract Loaded Ethylcellulose Microparticles as a Delivery System to Improve Olive Oils Oxidative Stability"

_resources, doi:10.3390/resources12010006_

Round 1

Reviewer 1 Report

The revision is attached in a separate document.

Author Response

Comment: Topic of manuscript is interesting and valuable and is within the thematic scope of the journal and special issue “Resource Extraction from Agricultural Products/Waste”. The authors conducted study on using olive mill pomace extract loaded ethylcellulose micoroparticles as antioxidant to stabilize/improve olive oils oxidative stability. The methods used to analyze the composition of the fruit are appropriate primary and secondary lipid oxidation products in samples contained different antioxidative additive and antioxidative potential are appropriate. However, there are some doubts and weaknesses which should be clarified and revised during revision process. My general and specific comments are given below:

Response: The authors would like to thank Reviewer 1 for taking into account the scientific quality of our manuscript. We appreciate the positive and constructive comments and suggestions on our manuscript.

Comment: The first weakness of presented refers to olive oils to which the antioxidative additive will be applied (extra-virgin, virgin). This is surprising and unreasonable because extra virgin and virgin olive oils naturally contain a wealth of antioxidant compounds. I think it is not allowed because these products have to be natural. All consumers in Europe believe in it. I agree that may would be some problems with blend or refined oils. The second problem with the advisability of using antioxidant additives into extra virgin and virgin olive oils in is the acid profile. Fatty acid profile in olive oil is dominated by the oleic acid C18:1 c9. This FA is more less vulnerable on oxidation than FAs belonged to PUFA. Thus olive oil is regarded as oxidative stabile comparing with many oils (rapeseed, safflower, corn etc. ). These refined oils contained different synthetic antioxidants (mainly based on tocopheryl molecule). Thus I recommend verification hypothesis and revision Introduction part. Please verify hypothesis

Response: We greatly appreciate your comment. We agree with your point of view, really, extra virgin and virgin olive oils naturally contain a wealth of antioxidant compounds. The biological activities associated with the consumption of extra virgin and virgin olive oils have spread the habitual used of these products. However, these products are used not only by the end consumer as a flavoring and cooking fat but also in the food industries as ingredient in a wide range of applications. In this context, the use of olive mill pomace extract loaded ethylcellulose microparticles as a delivery system can improve olive oils oxidative stability and could be applied as an interesting alternative of antioxidative additive when fat has to be introduced as ingredient in functional food and beverages. In addition, most edible oils are chemically unstable and susceptible to oxidative deterioration, especially when exposed to oxygen, light, moisture and temperature. That oxidative degradation result in a loss of nutritional quality and development-undesired flavors, affecting shelf stability and sensory properties of the oil. The use of microparticles with antioxidant activities may be useful to retard lipid auto-oxidation and increase the range of applications.

We also agree with the reviewer comment about fatty acid (FA) and polyunsaturated fatty acids (PUFAs) “Fatty acid profile in olive oil is dominated by the oleic acid C18:1 c9. This FA is more less vulnerable on oxidation than FAs belonged to PUFA”. In the last years, the oil industry has to pay special attention in this context, as oils, fats and fatty foods suffer stability problems. The oils with higher contents of unsaturated fatty acids, especially polyunsaturated fatty acids, are more susceptible to oxidation. In order to overcome the stability problems of oils and fats, synthetic antioxidants, such as butylated hydroxyanisole, butylated hydroxytoluene (BHT), tert-butyl hydroquinone have been used as food additives (despite the inherent problems).

Therefore, the purpose of this study was to evaluate the effectiveness of encapsulated olive mill pomace extract on the retarding lipidic oxidation of three types of olive oil, extra-virgin, virgin, and a blend of refined and virgin olive oils, compared to embedment in lipidic matrices of only olive mill pomace extract and synthetic antioxidants (BHA and BHT considered alone or in a blend) using the Schaal oven test.

The current study brings new insights into the design of natural antioxidants-rich loaded microparticles that be further incorporated in oils to retard oxidation processes occurring in the products that contain oils. The design of antioxidants-rich loaded microparticles may efficiently improve oils oxidative stability.

We've made improvements to the hypothesis and introduction section in the revised manuscript (Lines 41, 46 and 83).

Comment: Materials and methods: Part 2.2: Please supply more information about using solvents for extraction phenolic compounds. This is important for the quality of the manuscript (for future citations).

Response: Thank for the observation. The information was added in the revised manuscript (Line 127).

Comment: Part 2.4: Please verify appropriateness sentence in lines 182-184 accordingly: European Commission, Additives of the European Parliament and of the Council on food additives by establishing a Union 705 list of food additives approved for use in food additives, food enzymes, food flavourings and nutrients. Offical Journal of the 706 European Union 2011, 1130, 178–204

Response: Thanks for the information. The sentence “The enrichment of olive oil with BHA and BHT was performed up to the legal limit of 200 mg/kg of oil [19]. The legal limit of 200 mg/kg of oil - based on the amount of antioxidants - was also considered during the addition of the extract and microparticles to olive oils” was changed by “The enrichment of olive oil with BHA and BHT was performed up to the legal limit of 200 mg/kg of oil [19]. This reference value was considered during the addition of the extract and microparticles for olive oils fortification” (Line 197).

Comment: In part 2.5 please avoid discussion about methods. For example, the sentences in part 2.5.1 in Lines 208-215 should be move to R & D part. The same for the 2.5.2 part, please consider insert lines 227-232 in R & D part. Also please verify description of other methods

Response: Thank you for the observation. The manuscript was resided accordingly (Lines 337 and 395).

Comment: Results and Discussion: In Table 2 please consider to use whole names instead acronyms (is difficult to follow).

Response: We appreciate this observation. The manuscript was revised accordingly (Line 320).

Comment: There is not statistical differences between studied samples presented in Tables and Figures. Please supply.

Response: Thank you for the observation. Differences between means were analyzed by Tukey's test at a significance level of p ≤ 0.05, using SAS (version 9.3) software (Line 305). The statistical differences between studied samples are presented in the Tables (3 and 4) and Figure 5 (Lines 350, 438, and 577). The letters that allow comparing the means analyzed by Tukey's test were only added in the Tables where it is necessary to compare the means obtained from different samples. For Figures 1 to 4, the placement of the letters would not be suitable, so even though it was not presented, we discussed the results through verification of analysis of significant differences through One—way analysis of variance (one-way ANOVA), where values of  were considered statistically significant. All the corrections were made in the revised manuscript (Lines 305, 350, 438 and 577).

Comment: The results presented on Figures 3 and 4 are invisible. Please consider to change into one or two Tables.

Response: Thanks for the observation.  Through the two images, it is possible to see that some samples presented similar values on certain days. This is important to demonstrate behavior over the days. In addition, through the images, it is easier to visualize this behavior. Therefore, we decided to keep the Figures 3 and 4 and increase the image size, instead of presenting the results in the table, which could difficult the analysis (Lines 529 and 546).

Comment: Figure 5 is the best presentation of obtained results. However significant changes between studied variants were not presented. Please supply.

Response: Thank you for the observation. The Figure was changed and we include the statistical differences between studied samples based on the differences between means were analyzed by Tukey's test at a significance level of p ≤ 0.05, using SAS (version 9.3) software (Line 577).

Reviewer 2 Report

The work entitled "Olive mill pomace extract loaded ethylcellulose microparticles as a delivery system to improve olive oils oxidative stability" was well presented showing high scientific interest. The importance of using natural preservatives is clearly stated in the present paper. Based on this research ethylcellulose microparticles can act as an effective matrix for the encapsulation of phenolic compounds derived from olive mill pomace with positive effects on the shelf life of olive oil.

Author Response

Comment: The work entitled "Olive mill pomace extract loaded ethylcellulose microparticles as a delivery system to improve olive oils oxidative stability" was well presented showing high scientific interest. The importance of using natural preservatives is clearly stated in the present paper. Based on this research ethylcellulose microparticles can act as an effective matrix for the encapsulation of phenolic compounds derived from olive mill pomace with positive effects on the shelf life of olive oil.

Response: The authors would like to thank Reviewer 2 for taking into account the scientific quality of our manuscript and considering it worthy of publication.

Reviewer 3 Report

-In formula 1 on page 4, the variables are marked with capital letters, and in the text the same variables are marked with lower case letters, I suggest using either upper or lower case letters to mark the variables.

-add a reference to the next statement in the line 251 : "Nevertheless, long-chain fatty acids can be converted into short-chain fatty acids that may release free fatty acids (FFA) with time"

Author Response

Comment: In formula 1 on page 4, the variables are marked with capital letters, and in the text the same variables are marked with lower case letters, I suggest using either upper or lower case letters to mark the variables.

Response: Thank you for the observation. We opted to use lower case letters in the revised manuscript (Lines 169 and 183).

Comment: add a reference to the next statement in the line 251 : "Nevertheless, long-chain fatty acids can be converted into short-chain fatty acids that may release free fatty acids (FFA) with time"

Response: We appreciate this observation. The reference was added to the revised manuscript (Line 253).

Reviewer 4 Report

The overall aim of the research is  clearly presented;

The aims of the manuscript and the results of the data are clearly and concisely stated in the abstract.

The introduction provide sufficient background information to enable readers to better understand the problem being identified .

The Authors provided sufficient evidence for the claims that  they are making.

The data presented are of quality and has it been analyzed correctly

The figures and tables help the reader better understand the conclusions.

Author Response

Comment: The overall aim of the research is clearly presented; The aims of the manuscript and the results of the data are clearly and concisely stated in the abstract. The introduction provide sufficient background information to enable readers to better understand the problem being identified ; The Authors provided sufficient evidence for the claims that  they are making. The data presented are of quality and has it been analyzed correctly. The figures and tables help the reader better understand the conclusions.

Response: We appreciate these observations. The authors would like to thank Reviewer 4 for taking into account the scientific quality of our manuscript and considering it worthy of publication.

Round 2

Reviewer 1 Report

In my opinion the manuscript was sufficiently revised accordingly to the comments. I also appreciate responses on all comments.